# Gaps, traps, bridges and props: a mixed-methods study of resilience in the medicines management system for patients with heart failure at hospital discharge

Beth Fylan,[1,2] Iuri Marques,[1] Hanif Ismail,[1] Liz Breen,[1] Peter Gardner,[3] Gerry Armitage,[1,4] Alison Blenkinsopp,[1] On behalf of the ISCOMAT Programme Team

¹School of Pharmacy and Medical Sciences, University of Bradford, Bradford, UK
²Yorkshire Quality & Safety Research Group, Bradford Institute for Health Research, Bradford, UK
³School of Psychology, University of Leeds, Leeds, UK
⁴Bradford District Care Trust, Bradford, UK

**Correspondence to**
Dr Beth Fylan;
b.fylangwynn@bradford.ac.uk

## ABSTRACT

**Introduction** Poor medicines management places patients at risk, particularly during care transitions. For patients with heart failure (HF), optimal medicines management is crucial to control symptoms and prevent hospital readmission. This study explored the concept of resilience using HF as an example condition to understand how the system compensates for known and unknown weaknesses.

**Methods** We explored resilience using a mixed-methods approach in four healthcare economies in the north of England. Data from hospital site observations, healthcare staff and patient interviews, and documentary analysis were collected between June 2016 and March 2017. Data were synthesised and analysed using framework analysis.

**Results** Interviews were conducted with 45 healthcare professionals, with 20 patients at three time points and 189 hours of observation were undertaken. We identified four primary inter-related themes concerning organisational resilience. These were named as gaps, traps, bridges and props. Gaps were discontinuities in processes that had the potential to result in poorly optimised medicines. Traps were features of the system that could produce errors or unintended adverse medication events. Bridges were features of the medicines management system that promoted safety and continuity which ensured that, despite varying conditions, care could be delivered successfully. Props were informal, temporary or impromptu actions taken by patients or healthcare staff to avoid potential adverse events.

**Conclusion** The numerous opportunities for HF patient safety to be compromised and for suboptimal medicines management during this common care transition are mitigated by system resilience. Cross-organisational bridges and temporary fixes or 'props' put in place by patients and carers, healthcare teams and organisations are critical for safe and optimal care to be delivered in the face of continued system pressures.

## INTRODUCTION

The WHO views the safe management of medicines as a global challenge.[1] In the UK, guidance stresses the importance of improving the way medicines are managed at care transitions, such as admission to and discharge from hospital.[2] However, there is continued evidence that systems managing medicines are not optimally calibrated, particularly during and after hospital discharge,[3] when the responsibility for patient care shifts across organisations and clinicians.[4]

Medicines management is a system that supports the therapeutic use of medicines by patients, involving multiple healthcare organisations and staff with different clinical specialties and professional roles.[5] There is no shortage of evidence about the points at which healthcare systems fail to provide safe care.[6–9] Patients are not always well prepared to leave hospital and self-manage

their ongoing treatment.[10] The effective transfer of sufficient and accurate information between healthcare organisations remains inadequate in many cases[4] and is compounded by boundaries between care providers who may not always have access to the same information about patients' health. It is then unsurprising that discrepancies arise between medicines lists held by different care providers and patients.[3 11 12]

**Strengths and limitations of this study**

► Using four geographical areas and mixed methods, this study explored medicines management for people with heart failure at a time of considerable risk.
► Multiple viewpoints of patients and staff highlighted how the system is resilient in the face of pressure and weaknesses.
► The study presents a framework within which to explore and understand resilience in healthcare systems: that of 'bridges' and 'props' set against a backdrop of 'gaps' and 'traps'.
► The study collected multiple viewpoints but did not include the perspectives of local, regional and national policymakers.
► The patients and staff who agreed to be interviewed may have had particularly positive or negative experiences of the system, although their accounts were augmented and triangulated by first-hand independent observations.

Heart failure (HF) is a chronic condition affecting 900 000 individuals in the UK and is projected to rise significantly with an ageing population.[13] HF is a costly condition for the National Health Service (NHS) and is characterised by high rates of readmissions.[14] HF symptoms and disease progression can be controlled through well-managed medicines; however, guidelines for their use are not always applied and cardiology medicines can also cause harm, such as kidney injury, if they are not monitored.[15] Hence, the optimal management of medicines when leaving hospital is crucial to enhance quality of life, manage symptoms, prevent deterioration leading to hospital readmission and reduce mortality.[16]

Current thinking in patient safety has shifted focus from the deconstruction of events leading up to safety incidents (Safety I) to a more positive and proactive view of healthcare systems that identifies and values what goes right as well as pinpointing what goes wrong (Safety II).[17 18] Thus Safety II focuses on preventing error while accepting that there is variability in the delivery of healthcare, acknowledging that patients do not always experience harm as a consequence of their care. Instead of reacting when things go wrong, organisations proactively anticipate developments, negative as well as positive. It offers recognition of good performance in the face of uncertainty, valuing flexibility, adaptability, foresight and knowledge of how systems operate.[17 19] This in turn promotes a more dynamic attitude to performance through resilience which is the ability for a system and the individuals therein to adjust prior to, during or following changes or disturbances or in the face of ongoing, sustained pressure.[18 20–22]

This concept of resilience in healthcare has looked at specific points in the patient pathway such as handover of care between staff in one location such as a ward or performing specific roles.[19 23 24] Only one previous study has explored how patients can enhance resilience in medicines management at and after hospital discharge through anticipating discrepancies and taking remedial action.[25] No studies to date have explored resilience in medicines management at this care transition from multiple perspectives, including staff and patients and across different healthcare economies.

This study aimed to address this evidence gap by systematically investigating resilience in the medicines management system, using HF as an example condition. More specifically, the study was designed to understand how the system compensates for weaknesses and maximises opportunities in order to deliver safe yet optimal treatment. Its objectives were to explore the system of medicines management in multiple healthcare economies to highlight where resilience exists and identify where improvements to the system can be made to enhance resilience.

## METHODS

We used a mixed-methods design in four healthcare economies and their local primary care organisations (one comprising two hospitals and three comprising one hospital) in the north of England. Sites were selected to include university teaching hospitals and non-university teaching hospitals in different areas. Data from site observations, staff and patient qualitative interviews and documentary analysis (discharge letters and organisational and national policies) were collected between June 2016 and March 2017. NHS research ethics committee approval was sought and granted (16/NS/0018).

### Patient involvement

A patient researcher was a member of the research team advising on patient recruitment, data collection materials and information and consent forms. The research was overseen by a patient-led steering group including people with HF and carers.

### Data collection

#### Observations

Following ward-level consent, three experienced health researchers (BF, HI, IM) conducted a total of 189 hours of observations in five cardiology wards and one HF clinic. Structured observation schedules developed by the research team informed by previous work[26] were used to record observations. We observed medicines and ward rounds, preparation of information for discharge, patient discharges, as well as any other impromptu medicines-related activities. Unstructured, contemporaneous field notes were taken by the researchers.

#### Patient recruitment

A quota sample of four to six patients in each site was constructed, aiming for at least 16 complete data sets in total in the four areas. Patients were recruited during hospital admission by research nurses in consultation with ward staff. Patients were eligible for the study if they were aged 18 years or older, had capacity to consent and had been admitted to hospital with a diagnosis of HF with reduced ejection fraction (<45%) measured by an echocardiogram within the last 5 years. Patients also needed to present New York Heart Association Class III symptoms.[27] Research nurses approached eligible patients to introduce the study. Patients were then provided with a participant information leaflet and given the opportunity to ask questions about the study; they were given at least 4 hours to decide whether or not to take part.

#### Patient interviews

Patients' experiences with their medicines were explored at three time points: at, or as soon as practicable, after discharge (covering experience from admission to

discharge) and then approximately 2 and 6 weeks later. The research team developed a semistructured interview schedule built on previous work[26] and a review of relevant literature. The schedule comprised questions relating to patients' experiences with their medicines, and prompts and probes were used when relevant. Two researchers conducted the interviews (BF, HI). Interviews lasted up to 60 min, took place in patients' homes and were video or audio recorded and transcribed verbatim.

### Healthcare staff recruitment
Healthcare professionals with a role in medicines management in primary or secondary care were approached to take part in a semistructured interview either by research nurses or the study team, using face-to-face communication or by email invitation. A range of healthcare professionals involved in medicines management were selected following ward observations.

### Healthcare staff interviews
An interview schedule was developed by the research team to explore staff perceptions of safe medicines management. The schedule focused on medicines management processes, staff views on its quality and effectiveness for patients with HF in primary and secondary care, and their experiences of medicines management at discharge from secondary to primary care. Staff were given a participant information leaflet describing the study and, if they agreed to take part, an appointment was made to conduct the interview. Interviews lasted up to 60 min, were audio recorded following written consent and transcribed verbatim.

### Analysis of key documents
Documents were identified and reviewed including: national guidance on medicines optimisation used in the hospital setting[2]; local policies on medicines management and discharge in the four health economies; case notes and communications such as discharge letters, and any patient information about medicines in use in the four hospitals and available as text. Examples of potential system resilience at care transitions and risks in the system were identified and using a framework that mapped them according to the point in the transition to which they related and to the resilience element (or lack of) they evidenced.[19]

### Data analysis
The process of data analysis was iterative and comparative: analysing the first round of interview and observation data as further interviews and observations were undertaken; providing the opportunity to explore emerging themes in greater detail in subsequent fieldwork. The research team met several times both during and following data collection to discuss the data synthesis and analysis method and the emerging themes. Interview data were synthesised through data extraction with the data from observations and documents and the combined data were analysed using the framework approach,[28] involving detailed

**Table 1** The number of patients interviewed at each time point by site

| Site | Time point 1 | Time point 2 | Time point 3 |
|------|--------------|--------------|--------------|
| Site 1 | 2 | 3 | 3 |
| Site 2 | 5 | 4 | 4 |
| Site 3 | 6 | 6 | 5 |
| Site 4 | 6 | 6 | 5 |
| Total | 19 | 19 | 17 |

familiarisation with the data, identifying themes, interpreting the findings within the context of similar research studies and considering policy and practice.

## RESULTS
A total of 55 interviews with 20 patients with HF were conducted: 19 at discharge or shortly afterwards (time point 1); 19 approximately 2 weeks after discharge (time point 2); and 17 approximately 6 weeks after discharge (time point 3). We were unable to contact one patient from site 1 at time point 1; at site 2 one patient withdrew from the study after the first interview. One patient from site 3 was not interviewed at the third time point due to hospital readmission, and at site 4 one patient was too ill to continue after the second interview. Table 1 presents the number of patients interviewed for each site at the different time points. Table 2 outlines the gender and age of interviewed patients.

Forty-five interviews (table 3) were conducted with healthcare professionals: 19 with primary care staff (15 in four general practitioner (GP) surgeries, two community pharmacists and two community HF nurses) and 26 with secondary care staff. Table 4 presents the number of healthcare staff interviewed by site.

We identified four primary inter-related themes concerning organisational resilience and termed these: gaps, traps, bridges and props. Examples representing each theme are shown in tables 5–8.

'*Gaps*' were defined as a discontinuity in key processes that form the medicines management system and had the potential to result in poorly optimised medicines.

**Table 2** The gender and age of patients who took part in interviews

| Site | Gender | Total | Age range |
|------|--------|-------|-----------|
| Site 1 | Male | 2 | 72–82 |
| | Female | 1 | 53 |
| Site 2 | Male | 5 | 40–89 |
| | Female | 0 | 0 |
| Site 3 | Male | 5 | 46–79 |
| | Female | 1 | 69 |
| Site 4 | Male | 4 | 46–78 |
| | Female | 2 | 69–76 |

**Table 3** Number of interviews by healthcare staff type

| Staff type | Number of interviews |
|---|---|
| GPs | 4 |
| Practice administrators/data quality managers | 2 |
| Practice pharmacists | 3 |
| Practice nurses | 1 |
| Practice managers | 3 |
| Community pharmacists | 2 |
| Community heart failure nurses | 2 |
| Clinical care coordinators | 1 |
| Community cardiac nurses | 1 |
| Cardiologists | 3 |
| Ward managers | 5 |
| Staff nurses | 2 |
| Junior sisters | 1 |
| Ward pharmacists | 3 |
| Specialist cardiology pharmacists | 2 |
| Consultant pharmacists | 1 |
| Junior doctors | 2 |
| Specialist heart failure nurses | 3 |
| Ward administrative staff | 4 |
| Total | 45 |

GP, general practitioner.

Approaches to preparing discharge information varied across sites, with information sometimes being missed due to a lack of preparation time. Gaps were also evident in the information shared with primary care and in the preparation of patients to use their medicines. For the latter, we identified no standardised processes for informing patients about their medicines and, while hospital policies stipulated that patients should be informed, and gave details of the types of information patients should have, there was no guidance on optimal methods for informing patients about their medicines or training, so patients' experiences of receiving medicines were inconsistent and information was deficient for some.

Discussions with some nurses during observations revealed that while they were aware of policies in place

**Table 4** The number of healthcare staff interviewed per site

| Site | Primary/community care | Secondary care |
|---|---|---|
| Site 1 | 6 | 7 |
| Site 2 | 4 | 8 |
| Site 3 | 2 | 4 |
| Site 4 | 7 | 7 |
| Total | 19 | 26 |

on what aspects to cover when discussing medicines with patients at discharge, they did not follow them and often rushed these conversations. (Site 2, field notes from ward observations)

After discharge we found gaps in the continuity of care, for example, not all patients had a community pharmacy Medicines Use Review because pharmacies did not routinely receive information about the patients' medicines at discharge. Waiting times for specialist staff follow-up varied considerably and were sometimes lengthy, for example, waiting times for an appointment with community HF specialist nurses who would manage medicines titration was sometimes as long as 3 months after discharge.

We defined '*traps*' as features of the way the medicines management system was designed or managed that might produce medication errors defined as a 'failure in the treatment process that leads to, or has the potential to lead to, harm to the patient'[29 30] or unintended adverse medication events. These were evident in the coordination of discharges, for example, the pressure on ward staff to expedite discharges as quickly as possible appeared to impact on the effective preparation of discharge information and on educating patients about their discharge medicines.

If you're busy, you'll write less and I think that's just what happens on the job. If I know I've got time, I'll make sure I input as much detail as possible, but if you're busy you just don't have the time to do that, so you'll just really do short summaries and just include the bare essentials. (Site 1, FY1 doctor)

Staff preparing discharge information were often interrupted, could not always locate patients' notes and none reported receiving training about safe practices with medicines at discharge to primary care. We also found error traps after discharge, such as a lack of time and resources in GP surgeries to process discharge information. Finally, there was evidence that patients' lack of knowledge about the purpose of their medicines could potentially cause confusion particularly when the changes made in hospital led patients to have different supplies or multiple multicompartment 'compliance aid' tablet boxes. Like hospital staff, none of the primary care staff had received formal training about safe practices with medicines at discharge to primary care.

'*Bridges*' were identified as formalised features of the medicines management system that had been made permanent and promoted the safety and continuity of medicines management. They ensured that, despite varying conditions, care could be delivered successfully to patients with HF.

When preparing the 'To Take Home' medicines at discharge, ward staff wait for the pharmacist to come to the ward to check the patients' medicines lists and ensure these are accurate and any errors can be rectified. (Site 2, field notes from ward observations)

**Table 5** Gaps at and after hospital discharge

| | At discharge | After discharge |
|---|---|---|
| Gaps | Discussions about medicines at discharge can be rushed due to time pressures and workload. | Community pharmacy is not integrated into communication about discharge medicines. |
| | No standard process or guidance on how to hold discussions with patients about medicines. | Patients are not routinely referred to community pharmacy for follow-up support. |
| | Limited or no formal training about care transitions, preparing discharge summaries or patients to use medicines for all staff. | Limitations to the extent of shared IT systems between primary and secondary care and between surgeries and pharmacies. |
| | Processes for preparing patients to go home with medicines are linear but not streamlined, for example, multiple staff members need to input which causes delays. | Not all surgeries have a practice pharmacist to reconcile medicines. |
| | Discharge summary information is technical and uses jargon and abbreviations which are difficult for patients to understand. | Long waiting times to access community heart failure nurse services (up to 12 weeks). |
| | Inconsistency in level of detail in information written on discharge summary due to workload and healthcare staff knowledge of the patient. | Some patients perceive limitations in posthospital follow-up care, including difficulty in accessing services in primary care. |
| | Varying information offered to patients about follow-up appointments. | Patients are not fully aware of the roles and skills of primary care staff, particularly community pharmacists. |
| | Limited awareness among staff about policies in place for medicines management. | Some patients unable to devise effective strategies to self-manage medicines at home. |
| | Effectiveness of discharge is not critically appraised due to lack of feedback (unless the patient is readmitted or primary care staff make queries). | |

IT, information technology.

Bridges also included methods of communicating with primary care about treatment, for example, when hospitals sent an electronic copy of the patient's discharge summary to their GP. In this case, summaries were put together by multidisciplinary teams including junior doctors, nurses and pharmacists who would check and add information about medicines that would be useful to the primary care team. After discharge, two participating hospital trusts ran pharmacist-led titration clinics to ensure that medicine doses were optimised. Titration clinics also meant patients would be seen more quickly than if they had to see a consultant cardiologist. One cardiology pharmacist explained that the titration clinic ensured patients' medicines were adjusted as and when appropriate, in light of some GPs not feeling confident about changing them.

GP practices differed in how they processed discharge information. In one practice, administrative staff would review the discharge summary and forward actions to practice staff if medicines information needed to be changed. In another practice, this task was the responsibility of the GP, who would forward actions to practice staff and book any tests needed as a consequence of any changes in medicines occurring during hospital stay (ie, blood tests). One GP reported that his practice had re-engineered their processing of discharge summaries to include a multidisciplinary team comprising administrative staff, a practice-based pharmacist (whose post was created in response to recognised safety risks) and GPs, with the pharmacist taking responsibility for coordinating the process.

And so [processing discharge summaries] it was in-between surgeries, it was at the end of the day, so it was being fitted in rather than having allocated time, so naturally when it's being fitted in the process is a bit more rushed, you're more under pressure, maybe your concentration levels aren't there, so mistakes can be easily made. So as a practice we made the decision that just in terms of a workload thing and also patient safety and efficiency it would be worth investing in sort of pharmacy services. (GP, site 1)

Practice pharmacists perceived that their specialist knowledge improved as a consequence of being involved in the discharge process, while further expediting the safe management of medicines for patients after discharge. Some practice staff described having targets in place linked to time taken to process discharge summaries, with some practices prioritising processing driven by the risk of readmission. One data quality manager explained that they tried to process discharges within 24 hours of receiving information from the hospital, including reconciling medicines, but also explained that they had a maximum of a week to complete it.

**Table 6** Traps at and after hospital discharge

|  | At discharge | After discharge |
|---|---|---|
| Traps | Patient knowledge of medicines when they are discharged is limited. | Community pharmacy does not routinely receive copies of patients' discharge summaries so cannot correct or query new GP prescriptions. |
|  | There is pressure on ward staff to discharge patients and free-up beds. | Patients have an ongoing lack of knowledge of their medicines once home. |
|  | Variation in ward staffing levels and varying numbers of discharges to perform each day. | No formal training for surgery staff to process discharge information. |
|  | Use of several different IT systems in producing information for discharge. | Lack of time and resources in surgery to process discharge information. |
|  | Staff preparing patients for discharge and information about discharge medicines are interrupted. | Systems allow old prescriptions to be issued when medicines have changed. |
|  | Preparing information for discharge routinely left to junior members of staff who may not be familiar with the patient. | Dosages are monitored and changed by staff in different organisations. |
|  | Conversations about medicines with patients at discharge can be left to the last minute. | Trust in healthcare professionals may lead to a lack of patient critical appraisal of the condition and medicines. |
|  | Patients transferred to discharge lounges to await medicines face an extra transfer of care. | Changes in medicines lead to patients having conflicting medicines and multicompartment compliance aid boxes at home. |
|  |  | Varying levels of communication across care organisations results in extra burden to patient who has to fill in the gaps. |
|  |  | Varying information about medicines changes provided to primary care may lead to healthcare staff having to make decisions based on assumptions. |
|  |  | Healthcare professions may not accept treatment recommendation by other healthcare professionals (eg, GP not accepting recommendations made by Heart Failure Specialist Nurse). |

GP, general practitioner; IT, information technology.

So we have a week turnaround in order to get any meds reconciliation done. We generally get our electronic discharge normally within 24 hours of the patient being discharged, that would be scanned through the system that will then go to the doctor, the doctor will then forward it to me generally for coding and also to our practice pharmacist. (Data quality manager, site 2)

'*Props*' were informal, temporary or impromptu actions taken by patients or healthcare staff to avoid potential adverse events, such as medication errors. Props were sometimes developed in response to risks in the working environment, such as interruptions during medicines rounds.

During medicines rounds, nurses are frequently interrupted whilst sorting patients' medicines. One nurse observed also uses the strategy of signing the drug chart soon as one medicine is sorted into the plastic cup before moving to the next medicine. If there are interruptions, the nurse will know which medicines have already been sorted by looking at the drug chart. The nurse says it is a brilliant strategy to ensure accuracy and safety and cope with inevitable distractions and interruptions. (Site 1, field notes from ward observations)

Hospital staff told us that they suspected recommendations made by the hospital (eg, the uptitration of doses which is critical in HF) may not be acted on in primary care. Hence, staff created solutions to prevent a break in the ongoing treatment, giving patients an extra copy of their discharge letter to take to the GP. Some staff members described being cognisant of how discharge information can be difficult for patients to understand and would take extra time to explain the discharge summary and any abbreviations contained within it. One staff nurse at site 1 described having to make protected time to hold these discussions with patients, drawing curtains around the patients' beds to prevent any disruption. Some patients reported being discharged with an insufficient amount of medicines, leading them to seek community pharmacists help to provide them with emergency supplies until they could see a GP. Some patients also proactively provided the necessary links between community pharmacy, GPs and

**Table 7** Bridges at and after discharge

| | At discharge | After discharge |
|---|---|---|
| Bridges | Hospitals have established methods of communicating about patients' treatment with primary care. | Some trusts provide outpatient clinics where patients can receive intravenous fluids, thus avoiding them to need to be admitted to receive these medicines or speeding up discharges. |
| | Preparing discharge summaries and To Take Out (TTO) lists is a multidisciplinary task involving nurses and pharmacists. | GP practices have systems for acting on discharge information once it is received, although processes and times to process this information vary. |
| | Ward pharmacists can expedite well-managed discharge through proactively creating TTO lists. | Some practices have targets in place linked to time to process discharge information (eg, 24 hours from receiving this information). |
| | One trust routinely referred patients to community pharmacy for follow-up support with their medicines. | One practice pharmacist re-engineered the process for action on discharge information. |
| | All hospitals had policies for informing patients about their medicines. | Some practices use practice pharmacists to improve and expedite the processing of discharge information. |
| | Heart failure nursing staff attempted to see patients before their discharge to talk about their medicines to avoid having these conversations rushed at discharge. | Community pharmacy is sometimes able to perform postdischarge Medicines Use Review for patients. |
| | In two trusts, ward-based pharmacists would speak to patients about their medicines before discharge. | Two hospital trusts run pharmacist-led titration clinics to manage patients' medicines, meaning that patients can be seen and followed up quickly. |
| | Patients received written information about their medicines, with one trust providing an easy-to-understand medicines chart occasionally annotated by staff. | Some practices have ambulatory services. |
| | Patients are referred to specialist heart failure teams for follow-up. | Heart failure specialist nurses offer support services including medicines optimisation. |
| | | Some GP practices have systems to identify discharged patients with high risk of being readmitted so they can take preventative action. |

GP, general practitioner.

the hospital after discharge. For example, one patient called the community pharmacy to ask what information, if any, they had been provided with about his medicines. Another patient provided their GP practice with information about dose changes.

> So when I'd run out, I rang my GP and they were blissfully unaware of any changes to the amount, the receptionist had to take it down. She says 'well what was you on?' and I said 'well I was on one tablet a day and then they took me down to half, then they put me to one tablet a day again and now I'm on two tablets a day.' 'Two tablets?', this is the receptionist's questions. I says 'yeah, two tablets.' (Patient 05, site 4, interview 2)

Finally, community pharmacists stepped in to organise supplies for patients when something had gone wrong and the patient was unable to get the correct medicines.

## DISCUSSION

Notwithstanding a considerable body of research that has illuminated the sometimes alarming levels of preventable harm in healthcare systems and how this could be reduced,[31 32] this study suggests that there are opportunities to enhance the system that manages medicines across multiple organisations. The study also provides a positive perspective on the strategies developed and actions taken by healthcare organisations, their staff and patients to provide care successfully in the face of continued pressure and gaps that appear between and between organisations in this complex system.[33] The multiple perspectives of patients and multidisciplinary staff, independent observation of practice and documentary analysis collected through mixed methods allowed the possibility of triangulating data from multiple sources to offer a thorough description of a complex system. In doing so we also present a framework within which to explore and understand resilience in healthcare systems: that of 'bridges' and 'props' set against a backdrop of 'gaps' and 'traps'. Moreover, this study explores a whole healthcare system inclusive of its transitions, to reveal the context of that system. In contrast, previous studies, although adding to an understanding of system resilience, have examined these problems from solely a health professional perspective[23] or a patient perspective.[25]

**Table 8** Props at and after hospital discharge

| | At discharge | After discharge |
|---|---|---|
| Props | Some staff create their own checklists to follow discharge processes, such as using the discharge summary to tick off medicines. | Patients create their own lists of medicines, going online to seek more information. |
| | Staff occasionally give patients two copies of the discharge summary so that patients can give one to their GPs in case they do not receive it electronically. | Community pharmacists who have received a copy of the discharge summary use them to check against repeat prescriptions before dispensing. |
| | Staff make ad hoc queries to establish reasons for medicines changes which are unclear and undocumented so that they can be clear on the discharge summary. | Patients check medicines prescribed by their GPs against their discharge summary and/or take a copy when go see the GP or update them verbally. |
| | Staff will delay discharge to wait for relatives to arrive so that they can include them in conversations about medicines. | GP identifying potentially problematic changes in medicines occurring in hospital due to their enhanced knowledge of the patient. |
| | Ward pharmacists give advice to patients if they are concerned about patients getting confused, for example, advising them to return their old medicines to the pharmacy for disposal and only take the new ones. | GPs try to fill in patients' knowledge gaps about their medicines after discharge. |
| | Patients write additional information on the medicines' boxes or ask staff to write it so that they can better manage their medicines at home, for example, time to take medicines. | Community pharmacy provides emergency supply of medicines when patients are discharged from hospital without sufficient medicines. |
| | Patients are sometimes cognisant of how difficult it is for patients to understand their medicines and information provided at discharge, so they take extra time to hold these conversations. | Patients are given telephone numbers for heart failure nurses to contact them after discharge because waiting times to be seen by them are long. |
| | Staff draw curtains around the patients' beds when talking to them to ensure privacy and prevent interruptions. | Heart failure nurses can identify where patients make mistakes taking their medicines, for example, continuing to take discontinued medicines. |
| | Nurses resist instructions to send patients to discharge lounges as they feel the staff will not have specialist knowledge, and provide enhanced instructions to discharge lounge if over-ruled. | Heart failure nurses use the patients as a conduit for information to be exchanged between them and other healthcare professionals. |
| | Junior doctors query with pharmacist on ward if they need additional information about medicines. | Patients develop individual strategies and routines to adhere to medicines at home, for example, alarms, writing additional information in the discharge summary, storage systems, affixing discharge summaries on the fridge, and so on. |
| | | Some patients take all their medicines to community pharmacy after discharge, seeking information on which medicines they should continue to take and which should be discarded. |

GP, general practitioner.

Bridges and props either provided permanent system adaptations to potential gaps in care, or temporary fixes, usually implemented by individuals or small teams. Sometimes the props were put in place despite organisational pressure, for example, to discharge patients and free beds. We also draw out the dissonance between what healthcare professionals believe should happen and the reality of contemporary practice. This was clear from the differences between the recommendations for hospital discharge from national guidance and local policy, where, for example, patients must be fully informed about their medicines and any changes, and the overall discharge process—which in the settings observed, may lack depth and the necessary detail, or appear rushed. This was sometimes due to different local conditions, such as the number of discharges that needed to be completed in a day, but also to local policies that lacked sufficient detail and were not supported by staff training. Healthcare systems are complex and non-linear and the Safety II paradigm asserts that success and failures are products of the same variable system performance and that linear models of events such as medication errors cannot reflect the complexity of modern healthcare systems.[34] An enhanced view of the system using a Safety II lens allows healthcare organisations and policymakers to understand and close the gap between work as imagined versus work

as done.[35] This view also provides a better understanding of how policies and guidelines are actually interpreted and whether they are implemented in healthcare organisations by staff who adjust their performance to deliver care in a complex system.

Resilient systems are able to learn from their clinical experience (both positive and negative), adapt to it and respond to provide successful outcomes.[36 37] It was evident that staff were able to anticipate system vulnerabilities, for example, in the transfer of discharge information, and take compensatory adaptive action in the form of 'props'. As found by a previous study, patients also took remedial action, such as providing missing information about medicines changes to staff.[25] Resilient systems can monitor, learn and anticipate opportunities to improve. A better understanding and acceptance of the error traps in the system present healthcare organisations with the opportunity to learn about how the system operates, particularly when it is under pressure and presents a basis to improve. A better knowledge of gaps allows staff to anticipate where problems may occur and take action to avoid them. Props in the system are indicators of how flexible staff and teams are and healthcare systems can learn from the temporary fixes put in place and knowing where bridges have successfully joined up care can help systems learn and be better placed to innovate elsewhere. There are opportunities to learn from the 'ordinary performance adjustments' that staff undertake to better understand how to keep patients safe,[37] thereby formalising system props into bridges.

### Implications for policy and practice

Successive UK government-commissioned reports have highlighted how care systems have failed and how the actions—or inactions of those who lead or contribute to the system—have sometimes led to poor care and patient harm.[38–40] Policymakers should recognise the attempts made routinely by healthcare professionals and teams to learn from their clinical experience and apply this learning to increase system resilience by delivering safer care for patients despite disruptive conditions, such as disconnected communication systems, varying staffing levels and the underprovision of formal training, for example, in discharge and care transfers. Our study has shown that improvements to both the efficiency and safety of care could be gained through connecting the discrete information technology systems that operate within and between organisations. Additionally, community pharmacists often remain isolated from the patient pathway and are not routinely included in the communication between secondary care and primary care practice, creating additional risk for patients with HF who must obtain new supplies of critical medicines often within 1 or 2 weeks of being discharged.[4] Implementing systems that enable community pharmacists to know about medicines changes made during hospital admissions and thus to reconcile subsequent GP prescriptions would improve safety of medicines management, especially for patients with HF whose medicines are very commonly changed following a period of acute care. Local electronic systems do exist in a small number of areas to ensure that the dispensers of postdischarge medicines are fully informed about the medicines hospital clinicians intended patients should take so that they can reconcile those medicines and ensure accurate ongoing supplies.[41] Policymakers also have a duty to help disseminate and promote implementation of these local innovations—such as the transfer of discharge medicines information to all agents in the medicines management system—which minimise inherent risk.

Patients, if they so desire, should also be provided with the opportunity to gain in-depth knowledge of their medicines before leaving hospital (or afterwards if they prefer), in order to enhance their ability to self-manage and monitor their condition; such knowledge might also increase patients' vigilance, their capacity for error detection, and therefore to ask for prompt support if medication problems arise. Materials to support patients should be developed using codesign methods to maximise their acceptability and usability with both patients and healthcare staff.[42] Policymakers may also consider allowing patients to write to and share a personal health record to keep track of and flag problems they may have with their medicines, and share these with their healthcare teams and report them to their care providers.[43] This would in some measure help address the under-reporting of medication errors, particularly in primary care.[44]

We found that staff received little formal training in coordinating medicines management, including in completing discharge summaries, and there was little evidence of interprofessional or cross-pathway training. Such training may foster a care environment where clinical and administrative staff have a better appreciation of the impact of the care they provide on different parts of the system, and on different colleagues. For example, how inadequate information on a discharge summary can cause difficulties for primary care staff attempting to reconcile medicines. Additionally, in primary care, understanding that the processing of discharge information can impact on patients and community pharmacists who must take action to ensure the correct medicines are supplied. Interprofessional education has been found to yield positive outcomes in healthcare and may be especially helpful here, although more evidence for its effectiveness has been called for.[45]

### Implications for future research

The Safety I paradigm produced valuable ways of unearthing and visualising risks within systems and explaining causation when accidents occur.[32] In healthcare systems, Safety II can add substantially by focusing on how safe care is delivered in the face of disruption and pressure by way of bridges, props or both, from individual, micro (eg, healthcare teams) and macro (eg, organisational) perspectives.[19] Investigating further how this happens for different health conditions using tailored

methodologies will allow a better understanding of safe, resilient care, and afford commissioners a view of how changes to services may impact on a complex system.[46]

## Limitations

We observed practice in four NHS trusts and interviewed a wide range of healthcare staff across the pathway and patients, alongside reviewing key documents, we did not include the perspectives of local, regional and national policymakers, which may have enhanced the understanding of how systems are designed and the gaps between design and delivery. Nevertheless, we were able to collect a large amount of data to compare policy practice which enhanced reliability and validity. The patients and staff who agreed to be interviewed may have had particularly positive or negative experiences of the system, although their accounts were triangulated by first-hand independent observations. Finally, the study was conducted in four NHS healthcare economies, at a time of heightened focus on the quality of healthcare, and reports of unprecedented financial constraints, which may have impacted on people's perspectives of care received and delivered, and on the nature of the care observed.

## CONCLUSION

There are numerous opportunities for patient safety to be compromised and medicines to be suboptimally managed during this care transition. However, there are also cross-organisational bridges and temporary fixes in the form of props, put in place by individuals, including patients and carers, and teams to maximise the opportunity for safe and optimal care to be delivered. Investigating gaps and traps in the healthcare system and identifying existing compensatory props and bridges allow the illustration of areas where healthcare can be improved and fragmented communication minimised during care transitions.

**Acknowledgements** We thank the patients and NHS staff who took part in this research. We also thank other members of the ISCOMAT programme management team.

**Collaborators** Dr Sarah Alderson, Professor David Alldred, Professor Claire Hulme, Dr Ian Kellar, Professor Mohammed A Mohammed, Professor DK Theo Raynor, Dr Jon Silcock, Dr Roberta Longo, Mr Robert Turner, Professor John Wright.

**Contributors** BF drafted the protocol, developed data collection tools, conducted fieldwork, analysed the data and drafted the manuscript. IM conducted fieldwork, analysed the data and drafted the manuscript. HI conducted fieldwork, analysed the data and commented on the manuscript. LB advised on data analysis. PG drafted the protocol, directed the study, advised on data analysis and commented on the manuscript. GA designed the study, drafted the protocol, directed data analysis and drafted the manuscript. AB designed the study, drafted the protocol and edited the manuscript.

**Funding** This study was funded by the National Institute for Health Research Programme Grants for Applied Research 'Improving the safety and continuity of medicines management at care transitions' (ISCOMAT) RP-PG-0514-20009.

**Disclaimer** The views expressed are those of the authors and not necessarily those of the NHS, the NIHR or the Department of Health and Social Care.

**Competing interests** None declared.

**Patient consent** Not required.

**Ethics approval** North of Scotland Research Ethics Committee.

**Provenance and peer review** Not commissioned; externally peer reviewed.

**Data sharing statement** The transcripts of interviews are not available at the Dryad repository.

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
