## [Reviewer comments · BMJ Open]

ARTICLE DETAILS

TITLE (PROVISIONAL)	Gaps, traps, bridges and props: a mixed-methods study of resilience in the medicines management system for heart failure patients at hospital discharge
AUTHORS	Fylan, Beth; Marques, Iuri; Ismail, Hanif; Breen, Liz; Gardner, Peter; Armitage, Gerry; Blenkinsopp, Alison

VERSION 1 – REVIEW

REVIEWER	Gian Franco Gensini Position: Full Professor of Internal Medicine and Cardiology, Former Dean of the Faculty of Medicine, Florence., Director of CESMAV (centre for advanced medicine), Florence.
REVIEW RETURNED	06-May-2018

GENERAL COMMENTS	This is a well-written and real world rooted paper that aims to analyze the so-called resilience (expressed within a novel framework, as “bridges and props”) of individuals, teams and systems in the context of the challenging management of heart failure patients being discharged from hospital. The premise of the study is identifiable in the concept of patient safety based no longer and not only on the reconstruction of the actions that led to medical errors, but also, at the same time, on virtuous paths (Safety II vs Safety I). This premise highlights the complex relationships between patients and health system that are not only based on potential and factual errors, but rather on those positive and proactive elements that emerge in the midst of uncertainty, complexity, barriers, omissions, commissions, and are ultimately capable of driving a proper patient management. This study involved 45 health professionals and 20 patients examined in three distinct time points. The data used were diversified, that is collected from multiple perspectives and sources: interviews with healthcare staff and patients, but also clinical documents (discharge letters and organizational and national policies) and hospital site observations. However, it is not clear whether the staff and patients were blinded about the objective of the study: the lack of blindness could in fact affect the behavior (especially in a virtuous sense) of both groups, according to the Hawthorne effect, potentially generating a greater number of strategies to overcome difficulties, but at the same time raising the risk of underestimating the negative effect of gaps and traps.
---

	Furthermore, it should be emphasized that the study applies to a specific patient population, ie those suffering from systolic decompensation, that represent the minority of patients subject to readmissions due to heart failure. Most re-admissions due to decompensation take place in patients with preserved ejection fraction (pEF) and are often related to comorbidities. (Roger VL. Epidemiology of heart failure. Circ Res. 2013;113:646–59. doi:10.1161/CIRCRESAHA.113.300268.) Since RCTs performed up to now have not been able to identify treatments with a prognostic impact in patients with heart failure and pEF, in this specific group therapeutic uncertainty is greater than in patients with heart failure and reduced EF; since there are only a few strong recommendations in therapeutic guidelines about heart failure with pEF, the prescriptive variability is expected to be greater in this field, as greater is the probability of errors and/or changes in the management of medicines through the transition from hospital to primary care. Given that the presence of comorbidities further complicates the management of medical therapy, it would be interesting to know the distribution of comorbidities in the patients included in this study. With regard to bridges and props, the added-value of the titration clinics managed by pharmacists can be reinforced by the recent demonstration of the better arterial pressure control in hypertensive black people through pharmacological intervention led by pharmacists in barber shops versus the usual care. (N Engl J Med 2018; 378:1291-1301. DOI: 10.1056/NEJMoa1717250). In short, this study analyzes a delicate aspect of medical practice, namely the management of medical therapy at discharge from hospital. This is a complex and poly-parametric field, that has been correctly analyzed by the authors from multiple perspectives (hospital doctors, GPs, patients, pharmacists) and with an appropriate methodology. The study sheds light on some virtuous aspects of drug management at hospital discharge. In this sense it can be a basis (and/or an integration) for methodologically sound clinical studies (eg cluster RCTs). However, just because it is a case study, I consider mandatory to replace the term “demonstrates” with the term “suggests” (third line of the discussion). In short, I would accept the paper with minor revision.
--	--

REVIEWER	Mark Sujan University of Warwick, UK
REVIEW RETURNED	06-May-2018

GENERAL COMMENTS	Thank you for the opportunity to review your manuscript. The paper describes the application of a Safety-II framework to medicine management for patients with heart failure. As the authors point out, medicines management remains a patient safety concern, which has been reaffirmed recently with the publication of the report looking into medication errors in the NHS. The authors undertook a qualitative study, which is appropriate, and which is reported well. It is great to see an increasing interest in, and appreciation of Safety-II concepts in healthcare.
---

	I have a few suggestions for improvement, which are more around the definition and use of Safety-II concepts rather than specifically on the topic of medicines management. (1) terminology: it would be helpful to clarify key terms and concepts. For example p3/l42 "errors" - are these human errors, medication errors, or what kind of errors? p4/l3 "specific risks" and p15/l31 "minimise the risk of error" - again, how is risk defined? What does minimising the risk of error mean? Reducing the likelihood of its occurrence? (2) Safety-II: how do the concepts of error and risk sit with Safety-II? Error hasn't been defined in the paper, but is usually something like deviation from a procedure / standard, or a negative outcome. Risk could be defined as the combination of likelihood of occurrence of a hazard and the severity of the consequences. These are all negative. Safety-II does not normally operate with such concepts, so it would be helpful to include the authors' interpretation. (3) Novelty: I believe the novelty of the findings (in terms of Safety-II) might be slightly overstated. I would suggest consulting this now pretty much seminal reference (18 years old and still a great read): Cook, R.I., Render, M. and Woods, D.D., 2000. Gaps in the continuity of care and progress on patient safety. Bmj, 320(7237), pp.791-794. Since, there have been numerous publications that essentially cover the same ground of gaps, traps etc albeit with different vocabulary. The 3 books (soon 5) on Resilient Health Care might also be worth consulting, as well as the special issue on Resilience Engineering published in Reliability Engineering & System Safety (2015). (4) Learning: p15/l49 "Learn from failure" - I would argue this is a misrepresentation. Safety-II suggests organisations should learn from what goes right as well as from what goes wrong, with a focus on everyday clinical work. "Learn from failure" could be better called "Learn from experience". If you are interested in this specific topic, you might find this article of some use: Sujan, M.A., Huang, H. and Braithwaite, J., 2017. Learning from incidents in health care: Critique from a Safety-II perspective. Safety Science, 99, pp.115-121. (5) Accimap - this is somewhat sprung upon the reader. It should either be explained or not mentioned at all. (6) ResiMap - would require further clarification. How is this different from FRAM? What's the actual theoretical underpinning?
--	---

REVIEWER	Dr Duncan McNab NHS Education for Scotland, United Kingdom
REVIEW RETURNED	15-May-2018

GENERAL COMMENTS	Overall I think this is a really useful way to convey the principles of system resilience to frontline teams. Below are some specific comments for consideration and some general comments that I think would strengthen the discussion section. I think the discussion could include a fuller comparison with the existing resilience literature. ABSTRACT
---

	For gaps, traps, bridges and props - why only give an example of a trap. Seems to stick out not sure needed in abstract. INTRODUCTION Page 3 line 34 – does it follow that medicines need to be optimised – need another line re control by medicines but can cause harm?? METHODS Page 4 line 27 – 4 health economies – could this be clarified? There are 4 sites but 5 wards and one HF clinic – I think that this needs clarified – also why were these sites chosen – purposively sampled? Page 3 line 52 There is a more up to date definition of resilience In the fourth book (Resilience Engineering in Practice, 2010) the definition is given as, "The intrinsic ability of a system to adjust its functioning prior to, during, or following changes and disturbances, so that it can sustain required operations under both expected and unexpected conditions." This is the given on the website of the resilient healthcare network is, "A system is resilient if it can adjust its functioning prior to, during, or following events (changes, disturbances, and opportunities), and thereby sustain required operations under both expected and unexpected conditions." Page 4 line 18 - Resilience is therefore more than compensating for weaknesses but also responding to opportunities. I think you have studied this – for example by considering opportunities for patients to contribute to creating safety. Page 4 line 21 typo "where resilience in the exists" Page 5 line 6 - "A quota sample of between 16-24 admitted patients was constructed to allow for attrition, aiming for 16 complete datasets." For me this could be clearer. It could also state this is the composite total between the four sites. Page 5 line 38 – how were healthcare staff recruited – convenience sample/ purposive?? Page 6 line 6 - document analysis on page 4 line 30 stated that using case note review for document analysis, This is not mentioned here in the analysis section. Page 6 line 10 – states "Examples of system resilience at care transitions and risks in the system were extracted using a framework that mapped them according to the point in the transition to which they related and to the resilience element (or lack of) they evidenced." Are these examples of resilience potential recorded in official documents? Or does this refer to patient case note review and examples of resilience found? I don't think you can say a document (protocol/guidance etc) shows system is resilient – maybe it indicates that there is potential to be resilient. Page 6 line 24. It states that data analysis was iterative then that "The research team met several times to discuss the data synthesis and analysis method and the emerging themes." I think this should be more specific – did the research team meet between data collections to discuss the data synthesis and analysis method and the emerging themes? Page 6 line 31 - The emerging analysis was thematic. Could this be clarified – my understanding of framework analysis is that thematic analysis is conducted as part of the process – so themes emerge not the analysis? Page 8 line 15 second time heading "Results" has appeared Page 8 line 30 – a 'gap' defined as a discontinuity of key process. The following is given as an example of a gap:
--	--

“For the latter, we identified no standardised processes for informing patients about their medicines and, while hospital policies stipulated that patients should be informed, and gave details of the types of information patients should have, there was no guidance on optimal methods for informing patients about their medicines or training in doing so.”

The ‘gap’ is that the patients did not receive the correct information. The lack of guidance is a contributory factor (probably one of many) in why this happens. The lack of guidance and training is certainly a gap in the ‘system-as-found’ but is it a discontinuity of a key process or does it just not exist? It may be that this actually is a trap as it relates to the way the system is designed – there is no guidance or training. On page 9 line 34 – you infer that a similar problem in primary care is a trap and not a gap.

Page 10 line 40 - GP staff says prioritise based on risk readmission –is this formal and training given? If so, then agree it is a bridge, if not it is a prop.

Props – informal resilient behaviour – but as soon as this becomes the formal system then do they become bridges?

TABLES

In tables – few abbreviations need expanded HCA and MCCA
HFSN TTO

Table 7 row 8 - What are ambulatory services?

Table 8 row 7 - patients or staff are cognisant??

DISCUSSION

Page 15 line 48 - definition of resilience is from Hollnagel 2006 – more up to date as above. For example - one of props is waiting for relatives to arrive – this is a way of responding to an opportunistic condition change and so newer definition of resilience appropriate.

Page 15 – I think that the comparison to literature should be expanded to describe how these findings relates to the literature around resilience in healthcare.

For example, how do gaps, traps, bridges and props relate to Hollnagel’s cornerstones of resilience – the ability to monitor, learn, respond and anticipate?

Does shared learning around gaps and traps help identify leading indicators of trouble? Does a shared understanding of bridges and props help individuals and teams to monitor, learn and respond?

By exploring these – can new problems be anticipated and responses considered?

Traps are the system conditions that lead to gaps. Bridges are ways organisations mitigate against problems from gaps and traps - need to learn from both bridges and props. This may help respond to difficult (unspecified) system conditions to continue successful operations. This requires understanding traps causing gaps and existing bridges and sharing props – perhaps to see if can bring work-as-imagined and work-as-done closer.

Bridges and props may indicate resilience potential – ways to monitor performance, anticipate problems and respond. By exploring and sharing these – can then learn – and so improve response of individuals/teams.

Page 16 line 5 – policymakers should understand resilient action – perhaps, but not to this level – this is very important for local teams to learn from. Understanding system gaps, traps, bridges and props and sharing this knowledge. Policymakers should appreciate

	the actions of individuals and teams to provide safe care despite conditions. Page 17 line 30 – the Functional Resonance Analysis Method is surely worth mentioning as it comes from the Safety-II and Resilience Engineering world and can help show relationships between functions within a system that may help demonstrate functions that bridge or prop gaps. I'm not sure proposing ResiMap is helpful as there is no detail of what it is. For me most of early resilience research focuses on props and it is great that some research into bridges/props show system level change to improve resilience Conclusions Not sure you need this sentence in the conclusion: For example, some GP surgeries have systems in place to ensure the timely and efficient processing of discharge information
--	---

VERSION 1 – AUTHOR RESPONSE

Reviewer comments to address	Response	Text change
Reviewer 1		
Furthermore, it should be emphasized that the study applies to a specific patient population, ie those suffering from systolic decompensation, that represent the minority of patients subject to readmissions due to heart failure. Most re-admissions due to decompensation take place in patients with preserved ejection fraction (pEF) and are often related to comorbidities. (Roger VL. Epidemiology of heart failure. Circ Res. 2013;113:646–59. doi:10.1161/CIRCRESAHA.113.300268.) Since RCTs performed up to now have not been able to identify treatments with a prognostic impact in patients with heart failure and pEF, in this specific group therapeutic uncertainty is greater than in patients with heart failure and reduced EF; since there are only a few strong recommendations in therapeutic guidelines about heart failure with pEF, the prescriptive variability is expected to be greater in this field, as greater is the probability of errors and/or changes in the management of medicines through the transition from hospital to primary care. Given that the presence of comorbidities further complicates the management of medical therapy, it would be interesting to know the distribution of comorbidities in the patients included in this study.	We thank the reviewer for this thoughtful comment. Our focus was on HrEF patients exactly because of the evidence base around the combination of medicines being effective and that for a range of reasons – including poorly optimised medicines - many of these patients are readmitted for HF reasons.	

With regard to bridges and props, the added-value of the titration clinics managed by pharmacists can be reinforced by the recent demonstration of the better arterial pressure control in hypertensive black people through pharmacological intervention led by pharmacists in barber shops versus the usual care. (N Engl J Med 2018; 378:1291-1301. DOI: 10.1056/NEJMoa1717250).	Thank you for this reference	
However, just because it is a case study, I consider mandatory to replace the term "demonstrates" with the term "suggests" (third line of the discussion).	Thank you we have amended this	Notwithstanding a very considerable body of research that has illuminated the sometimes alarming level of preventable harm in healthcare systems and how this could be reduced, this study suggests that patients still face safety threats through inadequate medicines management. The
Reviewer 2		
(1) Terminology: it would be helpful to clarify key terms and concepts. For example p3/l42 "errors" - are these human errors, medication errors, or what kind of errors? p4/l13 "specific risks" and p15/l31 "minimise the risk of error" - again, how is risk defined? What does minimising the risk of error mean? Reducing the likelihood of its occurrence?	We have addressed this point	We defined 'traps' as features of the way the medicines management system was designed or managed that might produce medication errors defined as a 'failure in the treatment process that leads to, or has the potential to lead to, harm to the patient' (Ferner & Aronson 2006) Sometimes the props, were put in place despite organisational pressure, for example to discharge patients and free beds..
(2) Safety-II: how do the concepts of error and risk sit with Safety-II? Error hasn't been defined in the paper, but is usually something like deviation from a procedure / standard, or a negative outcome. Risk could be defined as the combination of likelihood of occurrence of a hazard and the	We appreciate that we have not been clear on this matter and have improved the text accordingly.	We have made multiple text changes to enhance clarity

severity of the consequences. These are all negative. Safety-II does not normally operate with such concepts, so it would be helpful to include the authors' interpretation.	We believe that the language of Safety 1 can be used to understand where there are opportunities to improve system performance through understanding where variations in performance have produced less than optimal outcomes. In a recent article Rebecca Lawton points out that the system approach now characterised as safety 1 described in 'An organisation with a memory' encompasses both learning and resilience. See: Lawton R. It Ain't What You Do (But the Way That You Do It): Will Safety II Transform the Way We Do Patient Safety? Comment on "False Dawns and New Horizons in Patient Safety Research and Practice". Int J Health Policy Manag 2018 [online first]	
Novelty: I believe the novelty of the findings (in terms of Safety-II) might be slightly overstated. I would suggest consulting this now pretty much seminal reference (18 years old and still a great read): Cook, R.I., Render, M. and Woods, D.D., 2000. Gaps in the continuity of care and progress on patient safety. Bmj, 320(7237), pp.791-794. Since, there have been numerous publications that essentially cover the same ground of gaps, traps etc albeit with different vocabulary. The 3	The strength of our study lies in the use of perspectives from multiple sources (including key policy documents) to identify gaps, traps, props and bridges. Additionally, we discuss how the focus on bridges and props highlight the	We have made changes throughout the text to better reflect this.

books (soon 5) on Resilient Health Care might also be worth consulting, as well as the special issue on Resilience Engineering published in Reliability Engineering & System Safety (2015).	mechanisms in which the system ensures resilience for the safe delivery of care, despite pressures.	
(4) Learning: p15/l49 "Learn from failure" - I would argue this is a misrepresentation. Safety-II suggests organisations should learn from what goes right as well as from what goes wrong, with a focus on everyday clinical work. "Learn from failure" could be better called "Learn from experience". If you are interested in this specific topic, you might find this article of some use: Sujan, M.A., Huang, H. and Braithwaite, J., 2017. Learning from incidents in health care: Critique from a Safety-II perspective. Safety Science, 99, pp.115-121.	We have reframed our discussion to include the reference suggested and focus on learning from experience rather than failures.	Resilient systems are able to learn from their clinical experience (both positive and negative, positive and negative) adapt to it and respond to provide successful outcomes
(5) Accimap - this is somewhat sprung upon the reader. It should either be explained or not mentioned at all.	See comment below	
(6) ResiMap - would require further clarification. How is this different from FRAM? What's the actual theoretical underpinning?	We have revisited this section and revised it	
Reviewer 3		
ABSTRACT For gaps, traps, bridges and props - why only give an example of a trap. Seems to stick out not sure needed in abstract.	We have deleted this example from the abstract	
INTRODUCTION Page 3 line 34 – does it follow that medicines need to be optimised – need another line re control by medicines but can cause harm??	Heart failure symptoms and disease progression can be controlled through well managed medicines; however guidelines for their use are not always applied and medicines can also cause harm, such as kidney injury, if they are not monitored.	
Page 4 line 27 – 4 health economies – could this be clarified? There are 4 sites but 5 wards and one HF clinic – I think that this needs clarified – also why were these sites chosen – purposively sampled?	Added that one health economy comprised two hospitals, and three comprised one hospital and respective local	We used a mixed-methods design in four healthcare economies and their local primary care organisations (one comprising two hospitals

	primary care organisations	and three comprising one hospital) in the north of England. Sites were selected to include University teaching hospitals and non-University teaching hospitals in different areas.
Page 3 line 52 There is a more up to date definition of resilience In the fourth book (Resilience Engineering in Practice, 2010) the definition is given as, "The intrinsic ability of a system to adjust its functioning prior to, during, or following changes and disturbances, so that it can sustain required operations under both expected and unexpected conditions." This is the given on the website of the resilient healthcare network is, "A system is resilient if it can adjust its functioning prior to, during, or following events (changes, disturbances, and opportunities), and thereby sustain required operations under both expected and unexpected conditions."	Definition updated and references added	This in turn promotes a more dynamic attitude to performance through resilience which we define here as the ability for a system and the individuals therein to adjust prior to, during of following bounce back after any changes disruption or failure or disturbances or in the face of ongoing, sustained pressure
Page 4 line 18 - Resilience is therefore more than compensating for weaknesses but also responding to opportunities. I think you have studied this – for example by considering opportunities for patients to contribute to creating safety.		More specifically, the study was designed to understand how the system compensates for weaknesses and maximises opportunities in order to deliver safe yet optimal treatment.
Page 4 line 21 typo "where resilience in the exists"	We have corrected this typo	
Page 5 line 6 - "A quota sample of between 16-24 admitted patients was constructed to allow for attrition, aiming for 16 complete datasets." For me this could be clearer. It could also state this is the composite total between the four sites.		A quota sample of 4-6 patients in each site was constructed, aiming for at least 16 complete datasets in total in the four areas
Page 5 line 38 – how were healthcare staff recruited – convenience sample/ purposive??	Have reworded this sentence	A range of healthcare professionals involved in medicines management were selected following ward observations
Page 6 line 6 - document analysis on page 4 line 30 stated that using case note review		case notes and communications such as

for document analysis, This is not mentioned here in the analysis section.		discharge letters. We aimed to include a range of healthcare professionals involved in medicines management.
Page 6 line 10 – states “Examples of system resilience at care transitions and risks in the system were extracted using a framework that mapped them according to the point in the transition to which they related and to the resilience element (or lack of) they evidenced.” Are these examples of resilience potential recorded in official documents? Or does this refer to patient case note review and examples of resilience found? I don’t think you can say a document (protocol/guidance etc) shows system is resilient – maybe it indicates that there is potential to be resilient.	They were identified as potential sources of resilience	Examples of potential system resilience at care transitions and risks in the system were identified and
Page 6 line 24. It states that data analysis was iterative then that “The research team met several times to discuss the data synthesis and analysis method and the emerging themes.” I think this should be more specific – did the research team meet between data collections to discuss the data synthesis and analysis method and the emerging themes?	We have clarified this	The research team met several times both during and following data collection to discuss the data synthesis and analysis method and the emerging themes
Page 6 line 31 - The emerging analysis was thematic. Could this be clarified – my understanding of framework analysis is that thematic analysis is conducted as part of the process – so themes emerge not the analysis?	We have deleted this confusing line	
Page 8 line 15 second time heading “Results” has appeared	Thank you we have deleted this	
Page 8 line 30 – a ‘gap’ defined as a discontinuity of key process. The following is given as an example of a gap: “For the latter, we identified no standardised processes for informing patients about their medicines and, while hospital policies stipulated that patients should be informed, and gave details of the types of information patients should have, there was no guidance on optimal methods for informing patients about their medicines or training in doing so.” The ‘gap’ is that the patients did not receive the correct information. The lack of guidance is a contributory factor (probably one of many) in why this happens. The lack of guidance and training is certainly a gap in the ‘system-as-found’ but is it a discontinuity of a key process or does it just not exist? It may be that this actually is a trap as it relates to the way the system is	Thank you for pointing out this inconsistency.	For the latter, we identified no standardised processes for informing patients about their medicines and, while hospital policies stipulated that patients should be informed, and gave details of the types of information patients should have, there was no guidance on optimal methods for informing patients about their medicines or training, so patients’ experiences of receiving medicines were inconsistent and

designed – there is no guidance or training. On page 9 line 34 – you infer that a similar problem in primary care is a trap and not a gap.		information was deficient for some.
TABLES In tables – few abbreviations need expanded HCA and MCCA HFSN TTO Table 7 row 8 - What are ambulatory services? Table 8 row 7 - patients or staff are cognisant??	We have done this	
DISCUSSION Page 15 line 48 - definition of resilience is from Hollnagel 2006 – more up to date as above. For example - one of props is waiting for relatives to arrive – this is a way of responding to an opportunistic condition change and so newer definition of resilience appropriate.	We have amended the definition	
Page 15 – I think that the comparison to literature should be expanded to describe how these findings relates to the literature around resilience in healthcare. For example, how do gaps, traps, bridges and props relate to Hollnagel's cornerstones of resilience – the ability to monitor, learn, respond and anticipate? Does shared learning around gaps and traps help identify leading indicators of trouble? Does a shared understanding of bridges and props help individuals and teams to monitor, learn and respond? By exploring these – can new problems be anticipated and responses considered? Traps are the system conditions that lead to gaps. Bridges are ways organisations mitigate against problems from gaps and traps - need to learn from both bridges and props. This may help respond to difficult (unspecified) system conditions to continue successful operations. This requires understanding traps causing gaps and existing bridges and sharing props – perhaps to see if can bring work-as-imagined and work-as-done closer. Bridges and props may indicate resilience potential – ways to monitor performance, anticipate problems and respond. By exploring and sharing these – can then learn – and so improve response of individuals/teams.		Resilient systems can monitor, learn and anticipate opportunities to improve (REF Hollnagel). A better understanding and acceptance of the error traps in the system present healthcare organisations with the opportunity to learn about how the system operates, particularly when it is under pressure and presents a basis to improve. A better knowledge of gaps allows staff to anticipate where problems may occur and take action to avoid them. Props in the system are indicators of how flexible staff and teams are and healthcare systems can learn from the temporary fixes put in place and knowing where bridges have successfully joined up care can help systems learn and be better placed to innovate elsewhere.

		Work as imagined versus work as done An enhanced view of the system using this lens allows policy-makers to understand the gap between work as imagined versus work as done and better understand how policies and guidelines are actually implemented (or not) in healthcare organisations by staff who adjust their performance to deliver care in a complex system. This may serve to closer align work as imagined versus work as done.
Page 16 line 5 – policymakers should understand resilient action – perhaps, but not to this level – this is very important for local teams to learn from. Understanding system gaps, traps, bridges and props and sharing this knowledge. Policymakers should appreciate the actions of individuals and teams to provide safe care despite conditions.		Policymakers should recognise the attempts made routinely by healthcare professionals and healthcare teams to learn from their clinical experience and apply this learning to increase system resilience by delivering safer care for patients despite disruptive conditions, such as disconnected communication systems, varying staffing levels and the under-provision of formal training, for example in discharge and care transfers.
Page 17 line 30 – the Functional Resonance Analysis Method is surely worth mentioning as it comes from the Safety-II and Resilience Engineering world and can help show relationships between functions within a system that may help demonstrate functions that bridge or prop gaps. I’m not sure proposing ResiMap is helpful as there is no detail of what it is. For me most of early resilience research focuses on props and it is great that some	See section “Implications for future research”	

research into bridges/props show system level change to improve resilience		
Conclusions Not sure you need this sentence in the conclusion: For example, some GP surgeries have systems in place to ensure the timely and efficient processing of discharge information	We have removed this sentence	

VERSION 2 – REVIEW

REVIEWER	Mark Sujan University of Warwick, UK
REVIEW RETURNED	08-Jul-2018

GENERAL COMMENTS	The authors have addressed previous suggestions, and I believe the manuscript has improved. The paper makes a useful contribution towards highlighting the importance of studying everyday clinical work in order to better understand how healthcare organisations provide good quality care, and how they can improve further. A strength of the study is the large sample size (for a qualitative study), and the inclusion of a significant number of patient interviews.
---